



# A Budyko framework for estimating how spatial heterogeneity and lateral moisture redistribution affect average evapotranspiration rates as seen from the atmosphere

Elham Rouholahnejad [1], James W. Kirchner[1,2,3]

[1]Department of Environmental Systems Science, ETH Zurich, CH-8092 Zürich, Switzerland
[2]Swiss Federal Research Institute WSL, CH-8903 Birmensdorf, Switzerland
[3]Dept. of Earth and Planetary Science, University of California, Berkeley, CA 94720 USA

*Correspondence to*: Elham Rouholahnejad, elham.rouholahnejad@gmail.com

**Abstract.** Most earth system models are based on grid-averaged soil columns that do not communicate with one another, and that average over considerable sub-grid heterogeneity in land surface properties, precipitation (P), and potential evapotranspiration (PET). These models also typically ignore topographically driven lateral redistribution of water (either as groundwater or surface flows), both within and between model grid cells. Here we present a first attempt to quantify the effects of spatial heterogeneity and lateral redistribution on grid-cell-averaged evapotranspiration (ET) as seen from the atmosphere over heterogeneous landscapes. Our approach uses Budyko curves, as a simple model of ET as a function of atmospheric forcing by P and PET. From these Budyko curves, we derive a simple sub-grid closure relation that quantifies how spatial heterogeneity affects average ET as seen from the atmosphere. We show that averaging over sub-grid heterogeneity in P and PET, as typical earth system models do, leads to overestimates of average ET. For a sample high-relief grid cell in the Himalaya, this overestimation bias is shown to be roughly 12 %; for adjacent lower-relief grid cells it is substantially smaller. We use a similar approach to derive sub-grid closure relations that quantify how lateral redistribution of water could alter average ET as seen from the atmosphere. We derive expressions for the maximum possible effect of lateral redistribution on average ET, and the amount of lateral redistribution required to achieve this effect, using only estimates of P and PET in possible source and recipient locations as inputs. We show that where the aridity index P/PET increases with altitude, gravitationally driven lateral redistribution will increase average ET (and models that overlook lateral redistribution will underestimate average ET). Conversely, where the aridity index P/PET decreases with altitude, gravitationally driven lateral redistribution will decrease average ET. The effects of both sub-grid heterogeneity and lateral redistribution will be most pronounced where P is inversely correlated with PET across the landscape. Our analysis provides first-order estimates of the magnitudes of these sub-grid effects, as a guide for more detailed modeling and analysis.



## 1 Introduction

The atmosphere integrates the fluxes of water, energy, and trace gases that it receives from the spatially heterogeneous landscape beneath it. Earth system models typically account for this spatial heterogeneity, and the atmosphere's integration of it, only at scales larger than their relatively coarse grid resolution. Accounting for the considerable heterogeneity of the Earth's

surface at smaller scales, and its consequences for fluxes from the surface to the atmosphere, is a major challenge in Earth system modeling.

The grid resolution in earth system models is typically translated directly onto the Earth's surface, which is modeled as columns that are vertically disaggregated into soil layers at scales of centimeters or meters, but are horizontally averaged at the 1° by 1° (roughly 100 km by 100 km) scale of the overlying atmospheric model (Fig. 1). At this scale, individual ridges and valleys

disappear, and even major mountain ranges and basins can become indistinct. Likewise, much of the variability in the surface climatology of the landscape and its consequences for land-atmosphere interactions are lost.

This loss of detail in land surface properties has important implications for water fluxes in Earth system models. Given that ET may depend nonlinearly on both water availability and atmospheric water demand, which are both spatially variable at scales far below typical model grid scales, the average ET over a heterogeneous landscape may differ substantially from model

ET estimates derived from spatially averaged land surface properties. The potential importance of this issue has motivated research into methods for capturing sub-grid-scale properties and processes within Earth system models.

Nesting higher-resolution regional models within global models represents an obvious, but computationally demanding, approach to treating sub-grid scale heterogeneity. As described by Klink (, 1995), two broad classes of aggregation schemes have been proposed to incorporate sub-grid heterogeneity while keeping computational costs manageable. In "averaged"

surface schemes, the surface properties are averaged over each grid cell and the average is applied directly in the model. In "mosaic" schemes, by contrast, individual grid cells are partitioned into several surface types, the model is run for each surface type separately, and the fluxes from each surface type are area-weighted to determine the average fluxes for the grid cell.

Numerous modeling studies over the past two decades have shown that, in comparison to mosaic schemes and nested high-resolution models, averaged surface schemes tend to overestimate evapotranspiration and sensible heat flux (e.g., Klink, 1995;

Giorgi and Avissar, 1997; Essery et al., 2003; Teluguntala et al., 2011; Ershadi et al., 2013). Studies with nested high-resolution models demonstrate that this overestimation bias is largest where topographic effects play a major role (Giorgi and Avissar, 1997; Pope and Stratton, 2002; Boyle and Klein, 2010; Bacmeister et al., 2014)

Another potential source of bias in Earth system models arises from their neglect of surface and subsurface flows within and between grid cells. Current Earth system models calculate infiltration and vertical transport of water in each soil column, but

the water that reaches the bottom of the column is either stored as groundwater or simply disappears, reappearing later in the ocean. In real-world landscapes, by contrast, significant volumes of water are transported laterally, either via groundwater flow or by rivers flowing from mountains into valleys and potentially redistributing their water to valley ecosystems by infiltration into valley aquifers. These lateral redistribution processes supply water for evapotranspiration in groundwater-





dependent ecosystems in the dry season (Fan and Miguez-Macho, 2010). Several case studies in the Amazon (Christoffersen et al., 2014), central Argentina (Contreras et al., 2011; Jobbágy et al., 2011) and other groundwater-dependent ecosystems (Eamus et al., 2015) demonstrate how water supply can govern the seasonality and magnitude of evapotranspiration in those regions. However, the potential effects of these lateral redistribution processes on grid-scale ET, as viewed from the
atmosphere, are missing from current Earth system models, and the resulting biases in modeled water fluxes are unknown.

The Earth system modeling community has recognized the need to determine how sub-grid heterogeneity and lateral redistribution affect grid-scale evapotranspiration rates as viewed from the atmosphere, and to develop schemes that can efficiently account for these effects in land surface models (Clark et al., 2015). A recent high-resolution modeling study for the continental US (Maxwell and Condon, 2016) concluded that lateral redistribution could substantially alter the partitioning
of ET between transpiration and bare-soil evaporation, but the net effect on the combined ET flux remains unclear. The studies outlined above illustrate the potential effects of spatial heterogeneity and lateral redistribution, but we currently lack a general framework for estimating the resulting biases in calculated evapotranspiration rates. Here we present a first attempt to fill this knowledge gap, using an analysis based on Budyko curves as simple semi-empirical estimators of ET. This analysis yields first-order estimates of the potential effects of sub-grid heterogeneity and subsurface lateral redistribution on ET fluxes from
heterogeneous landscapes, as seen from the atmosphere.

## 2 A Budyko framework for estimating terrestrial water partitioning

The simplest widely used approach for estimating evapotranspiration rates from the land surface is the Budyko framework (Turc, 1954; Mezentsev, 1955; Pike, 1964; Budyko, 1974; Fu, 1981; Milly, 1993; Zhang et al., 2001; Yang et al., 2007). Budyko showed empirically that under steady-state conditions in catchments without significant groundwater inputs, losses or
storage changes, the long-term annual average evapotranspiration (ET) rate is functionally related to both the supply of moisture from the atmosphere (precipitation, P) and the evaporative demand for water by the atmosphere (potential evapotranspiration, PET). Under arid conditions (that is, when P is much smaller than PET), ET converges toward P, implying that ET is limited by the available supply of water (Fig. 2, water limit line). Alternatively, under humid conditions (that is, when P is much greater than PET), ET is limited by atmospheric demand and E converges toward PET (Fig. 2, energy limit
line). Budyko's original work showed, and decades of studies have confirmed, that under the long-term steady-state assumptions outlined above, hydrological systems typically operate close to either the energy or water constraints.

Several studies have explored how natural systems may violate the assumptions of the Budyko approach. Net inputs or losses of groundwater, as well as long-term changes in soil moisture and groundwater storage, have been shown to alter the water balance sufficiently that measurements of P and ET can produce points that fall far from the energy and water constraints in
Fig. 2. However, these apparent violations of the Budyko approach can be corrected if the precipitation term P is replaced by an effective precipitation that accounts for root zone water storage changes and net inputs or losses of groundwater (Zhang et





al., 2001; Zhang et al., 2008; O'Grady et al., 2011; Istanbulluoglu et al., 2012; Wang, 2012; Chen et al., 2013; Troch et al., 2013; Du et al., 2016).

The Budyko framework can be expressed in two different non-dimensional sets of axes, depending on whether one wishes to focus on the effects of changing water supply (P) or atmospheric water demand (PET). If one seeks to analyze the effects of

changing PET under a fixed P, it is most intuitive to non-dimensionalize both axes by P, as shown in Fig. 2a. In this coordinate space, translation along the horizontal axis represents a change in PET.

Our analysis, by contrast, focuses on how changes in water availability affect ET under a fixed PET. For such questions, it is most intuitive to non-dimensionalize the coordinate axes by PET, as shown in Fig. 2b. In this coordinate space, translation left or right along the horizontal axis represents changes in water availability. Thus, this coordinate space is better suited to

our analysis.

Table 1 presents several alternative empirical equations that have been proposed for "Budyko curves" relating ET to P and PET. Our analysis will be based on the Turc-Mezentsev equation (Eq. 1 in Table 1), because it is the most widely used of the alternatives shown here. However, the differences among these formulas are unimportant for the analysis presented below.

Here we use Budyko curves as simple models for how ET is controlled by the supply of available moisture (as represented by

P) and evaporative demand (as represented by PET). We could have used more complex ecohydrological models to estimate ET instead, at the cost of increased complexity and reduced transparency. However, any such models must obey the same energy and water constraints that shape the behavior of catchments in the Budyko framework, so we would not expect their behavior to deviate greatly from the Budyko curves that are analyzed here. Thus the Budyko curves that we analyze here can be considered as approximations to the behavior of these more complex models. They also have an important advantage for

our purposes, namely that they specify ET as an explicit function of its main drivers P and PET, allowing us to derive general analytical results that might otherwise be difficult to infer from sets of simulation results.

## 3 Effects of sub-grid heterogeneity on ET in a Budyko framework

The water and energy constraints that limit ET imply that ET is an intrinsically nonlinear function of P and PET. Under arid conditions (with P<<PET), ET will increase almost linearly with P, but as conditions become more humid and the supply of

moisture exceeds the energy available to evaporate it (P>>PET), the energy constraint will hold ET nearly constant as P increases. Conversely, under humid conditions, ET will scale almost linearly with evaporative demand (as expressed by PET), but as conditions become more arid and the supply of moisture becomes limiting, ET will be constrained by P and will become largely independent of PET.

As shown in Fig. 3, the nonlinear behavior of ET as a function of P and PET is also reflected in Budyko curves, particularly

near the transition between humid and arid conditions (P/PET close to 1). This nonlinear behavior has important implications for estimates of average ET in heterogeneous landscapes.





As Fig. 3 illustrates, the average of a nonlinear function with heterogeneous inputs will not, in general, be equal to the value of that function evaluated at the average of the input values. That is, the average of the function will not be the function of the average inputs (e.g., Rastetter et al., 1992; Giorgi and Avissar, 1997). One can visually see that the resulting heterogeneity bias will depend on how strongly curved the function is, and how widely its inputs are scattered. This intuitive concept can

be expressed mathematically by using the Method of Moments to estimate the heterogeneity bias (e.g., Ang and Tang, 1975). We begin by re-stating Eq. (1) from Table 1 as an explicit function of P and PET,

$$ ET = f(P, PET) = \frac{P}{\left( (\frac{P}{PET})^n + 1 \right)^{1/n}}. \quad (5) $$

For a function of two variables, a second-order, second-moment expansion leads directly to the following approximation for the mean of the function, in terms of the function's value at the mean of its inputs:

$$ \overline{ET} \approx f(\bar{P}, \overline{PET}) + \frac{1}{2} \frac{\partial^2 f}{\partial P^2} var(P) + \frac{1}{2} \frac{\partial^2 f}{\partial PET^2} var(PET) + \frac{\partial^2 f}{\partial P \, \partial PET} cov(P, PET) \quad , \quad (6) $$

where the derivatives are understood to be evaluated at $\bar{P}$ and $\overline{PET}$. Evaluating the necessary derivatives using Eq. (5) directly yields the following expression for the average ET,

$$ \overline{ET} \approx f(\bar{P}, \overline{PET}) - (n+1) \frac{\bar{P}^{n+1} \overline{PET}^{n+1}}{(\bar{P}^n + \overline{PET}^n)^{2+1/n}} \left[ \frac{1}{2} \frac{var(P)}{\bar{P}^2} + \frac{1}{2} \frac{var(PET)}{\overline{PET}^2} - \frac{cov(P, PET)}{\bar{P} \, \overline{PET}} \right] \quad , \quad (7) $$

where the second term represents the heterogeneity bias (that is, the difference between the average of the function and the function of the average). The relative magnitude of this bias can be derived by combining Eqs. (7) and (5), yielding

$$ \frac{f(\bar{P}, \overline{PET}) - \overline{ET}}{f(\bar{P}, \overline{PET})} \approx \frac{(n+1)}{\left\{ \left( \frac{\bar{P}}{\overline{PET}} \right)^{\frac{n}{2}} + \left( \frac{\overline{PET}}{\bar{P}} \right)^{\frac{n}{2}} \right\}^2} \left[ \frac{1}{2} \frac{var(P)}{\bar{P}^2} + \frac{1}{2} \frac{var(PET)}{\overline{PET}^2} - \frac{cov(P, PET)}{\bar{P} \, \overline{PET}} \right] \quad . \quad (8) $$

From Eq. (8) one can directly see that the heterogeneity bias will depend on the variances of P and PET, as well as their covariance (all nondimensionalized by their means). One can see that the heterogeneity bias will generally be positive (that is, estimates based on $\bar{P}$ and $\overline{PET}$ will overestimate $\overline{ET}$), because the covariance term in Eq. (8) will be less than the variance terms. One can also see that, all else equal, a negative correlation between P and PET will amplify the heterogeneity bias



(because, in terms of the Budyko plot, this will lead to greater scatter in P/PET). Furthermore, one can see that the relative heterogeneity bias will be greatest when the term in curly brackets in Eq. (8) will be as small as possible, which will occur at $\bar{P}/\overline{PET} = 1$ (the point of maximum curvature in the Budyko curve). Finally, from Eq. (8) one can see that at higher values of $n$, the peak heterogeneity bias will be greater (due to the $n+1$ term), but will be more tightly focused around $\bar{P}/\overline{PET} = 1$ (due

to the powers of $n/2$).

To estimate the heterogeneity bias that could result from grid-scale averaging in Earth system models, we applied the analysis outlined above to a 1° by 1° grid cell spanning the Himalayan Front in west Bhutan (Fig. 4a). This grid cell spans a sharp north-south topographic gradient, with altitudes ranging between ~500 m and ~6500 m. Within this grid cell, we compiled 30-arc-second values of P (WorldClim, Hijmans et al., 2005) and PET (MODIS, Mu et al., 2007) to examine the finer-scale

climatic drivers of variations in ET. Because 30 arc-seconds is approximately 1 km, we will refer to these as 1-km values for simplicity. One-kilometer P and PET, as well as 1-km values of ET estimated from these P and PET data using the Budyko curve (Eq. 5), vary strongly in this 1° by 1° grid cell, as shown in Figs. 4b, c, and d. The averages of these P, PET, and ET values over the 1° by 1° grid cell will plot as the yellow circle in Fig. 4e, lying well below the Budyko curve of the individual 1-km ET estimates. If instead we estimated the average ET for the grid cell from its average P and PET, we would obtain the

orange circle on the Budyko curve, corresponding to an 11.8 % overestimate of the true average of the 1-km ET values.

We repeated the same procedure to estimate the averaging bias in the 8 grid cells surrounding the one analyzed above (Fig. 5a). A comparison of these 9 grid cells shows that the averaging error is largest (around 13 %) when the variability in the aridity index (AI=P/PET), driven in turn by topographic variability, is largest (Fig. 5b,c,d).

## 4 Lateral redistribution by surface and subsurface flow, and its effects on average ET in a Budyko framework

Consider, as a thought experiment, an arid valley surrounded by high mountains. Evapotranspiration in the valley may depend not only on local precipitation in the valley, but also on precipitation that falls in the mountains and reaches the valley either by groundwater flow or by streamflow that re-infiltrates into valley aquifers. The lateral transfer of water from the mountains to the valley could clearly increase evapotranspiration rates in the valley by making more water available for vegetation, but could simultaneously make less water available for transpiration in the mountains. Will the net effect of this lateral transfer

be to increase, or decrease, average ET as seen from the atmosphere?

The mountains, the valley, and the lateral transfer between them will all be invisible at the grid scale of typical Earth system models. But the simple scenario described above suggests that lateral transport could alter the average ET over a model grid cell that incorporated both the mountains and the valley. What properties of the landscape will control the sign and the magnitude of the net effect on average ET? Here we extend the Budyko analysis presented above to estimate the potential

effects of lateral redistribution on average ET as seen from the atmosphere.

Our first step is to re-define the aridity index in the Budyko framework to take account of water that becomes available for evapotranspiration either through local precipitation or through net lateral transfer. In taking this step, we are implicitly



assuming that all water supplied to the ecosystem, from any source, is equally available for evapotranspiration. We introduce the term *available water* (AW), defined as

$$AW = P + net\ transfer \quad , \quad (9)$$

where *net transfer* represents the net influx of groundwater and re-infiltrating streamflow. Substituting available water for precipitation in the Turc-Mezentsev formula for the Budyko curve (Eq. 5), we obtain

$$ET = \frac{AW}{\left(\left(\frac{AW}{PET}\right)^n + 1\right)^{1/n}} = \frac{P + net\ transfer}{\left(\left(\frac{P + net\ transfer}{PET}\right)^n + 1\right)^{1/n}} \quad , \quad (10)$$

where AW is available water and, as before, ET is actual evapotranspiration, P, is precipitation, PET is potential evaporation, and *n* (dimensionless) is a catchment-specific parameter that modifies the partitioning of P between E and Q. Our approach follows the lead of several other investigators (Istanbulluoglu et al., 2012; Wang, 2012; Chen et al., 2013; Du et al., 2016) who have expanded the "precipitation" term to account for other sources of water in addition to precipitation per se. This approach assumes that lateral transfer alters only the available water at the two locations, and not their PET's.

**4.1 Two-column model and lateral transfer in Budyko space: graphical interpretation of the concept**

To continue the thought experiment outlined above, the mountain and valley environments described above could be represented by two columns of a land surface model, as shown in Fig. 6. Column 1 (the mountains, say) is a "source" column for lateral transfer of available water to Column 2 (the valley, say), which can be considered as a "recipient" column for this available water. In the example shown in Fig. 6, Column 1 has higher P and/or lower PET than Column 2. Laterally
transferring water from Column 1 to Column 2 will increase the water available for evapotranspiration (and thus ET itself) in Column 2, and will reduce them in Column 1. But will the increase in ET in Column 2 outweigh the decrease in ET in Column 1? That is, will the average ET as seen from the atmosphere increase or decrease, and by how much?

We can graphically illustrate the effects of lateral redistribution between the two columns in the Budyko framework as shown in Fig. 6b. The average ET of Column 1 and Column 2 will always lie on the line connecting the corresponding points on the
Budyko plot (and thus below the Budyko curve itself). As Fig. 6b shows, if we laterally transfer water from a more humid column to a more arid column, the corresponding points on the Budyko plot must move closer together, and the resulting average ET must move upward. Conversely, if we laterally transfer water from a more arid column to a more humid one, the corresponding points on the Budyko plot must move farther apart, and the average ET must decrease.

Because lateral transfer will necessarily be driven by gravity (and thus "source" locations will always lie above "recipient"
locations), the analysis shown in Fig. 6b leads directly to a simple general rule: wherever higher locations are more humid,





one should expect lateral redistribution to result in a net increase in ET, and conversely, wherever higher locations are more arid, lateral redistribution should result in a net decrease in ET.

As one can see from the graphical analysis shown in Figs. 6 and 7, the magnitude of the net ET effect will depend primarily on the amount of lateral redistribution (how far the points move along the Budyko curve) and on the degree of curvature

between them (and thus the _angle_ between the trajectories of the individual points). As shown in Fig. 7, if both locations are humid (and thus energy-limited) or both locations are arid (and thus water-limited), lateral transfer from one site to the other will have only a minimal effect on the average ET. If both sites are energy limited (and remain energy-limited), neither will respond strongly to a change in the amount of water available for evapotranspiration. If both sites are water limited (and remain water-limited), they will be almost equally sensitive to changes in available water; thus the increases in available water

and ET at one site will be nearly offset by the corresponding reductions at the other site. But if one site is water-limited and the other is energy-limited, then the responses of the two sites to changes in available water will be markedly different, and lateral transfer from one to the other could substantially affect the average ET over the two sites.

We emphasize that the analysis presented here is hypothetical. We are not asserting that lateral transfer actually occurs between the two columns, or even that it can occur between them, let alone what the magnitude of that lateral transfer is. Instead, we

are asking the hypothetical question: if water flows from one column to the other, how much would we expect the average ET to change, for each mm yr$^{-1}$ of water that is lost from one column and gained by the other?

### 4.2 Quantifying the effect of lateral transfer on average ET

We can make a first-order estimate of the net effect on ET using the Budyko curve as a simple model of ET rates. An illustrative calculation, for an extreme hypothetical case, is shown in Fig. 8. Column 1 is humid, with 2000 mm yr$^{-1}$ of annual precipitation

and a PET of 1000 mm yr$^{-1}$ (AI of 2.0), and Column 2 is arid, with 300 mm yr$^{-1}$ of annual precipitation and a PET of 2000 mm yr$^{-1}$ (AI of 0.15). As Fig. 8b shows, laterally transferring 200 mm yr$^{-1}$ from Column 1 to Column 2 would increase average ET by about 85 mm yr$^{-1}$, or 14 %.

We can generalize from this specific example by using Eq. (10) to calculate the average ET as a function of the amount of available water that is transferred from one column to the other,

$$ET_{avg} = 0.5 \left( \frac{(P_1 - x) \times PET_1}{((P_1 - x)^n + PET_1{}^n)^{\frac{1}{n}}} + \frac{(P_2 + x) \times PET_2}{((P_2 + x)^n + PET_2{}^n)^{\frac{1}{n}}} \right) \quad , \quad (11)$$

where $x$ represents the net transfer from one column to the other. Fig. 9 depicts how the average ET and the AW/PET and ET/PET ratios of the two sites change with lateral transfer. The average ET of the two columns increases with increasing net

transfer ($x$) up to a point, and then decreases for higher values of $x$. One can see from Fig. 9 that average ET reaches its





maximum when $x$ equalizes AW/PET (and thus ET/PET) at the two sites (note that this does not imply that either AW or PET are necessarily the same at the two sites).

We can verify this intuitive result by differentiating Eq. (11) by $x$:

$$\frac{dET_{avg}}{dx} = \frac{1}{\left(\left(\frac{P_2+x}{PET_2}\right)^n + 1\right)^{1+1/n}} - \frac{1}{\left(\left(\frac{P_1-x}{PET_1}\right)^n + 1\right)^{1+1/n}} \quad . \quad (12)$$

At the maximum $ET_{avg}$, $dET_{avg}/dx$ must equal zero, which can only occur if $x_{opt}$, the ET-maximizing rate of lateral transfer, is such that the two terms in Eq. (12) are equal, implying that:

$$\left(\frac{P_1-x_{opt}}{PET_1}\right)^n + 1 = \left(\frac{P_2+x_{opt}}{PET_2}\right)^n + 1 \rightarrow \frac{P_1-x_{opt}}{PET_1} = \frac{P_2+x_{opt}}{PET_2} = \frac{P_1+P_2}{PET_1+PET_2} \quad , \quad (13)$$

which shows directly that $AW/PET = (P \pm x_{opt})/PET$ in the two columns must be equal, confirming the intuitive result from Fig. 9. One can solve Eq. (13) to show that $x_{opt}$ will be the harmonic difference between $P_1$ and $P_2$, weighted by the reciprocals of the corresponding PET's:

$$x_{opt} = \frac{P_1 PET_2 - P_2 PET_1}{PET_1 + PET_2} = \frac{P_1/PET_1 - P_2/PET_2}{1/PET_1 + 1/PET_2} \quad . \quad (14)$$

The key result here (namely that ET is maximized when lateral transfer equalizes the ratio $AW/PET$ in the columns) is not restricted to two columns, and is not specific to the particular curve that we have analyzed here. Instead, it can be shown to be true for any downward-curving function on a Budyko plot, and for any number of interacting columns; for details see the Appendix.

The dimensionless quantity $dET_{avg}/dx$ (Eq. 12) expresses the change in average ET per unit of lateral redistribution. One quantity of particular interest could be the relative change in ET resulting from the first unit of lateral transfer, which can be obtained directly from Eq. (12) with $x = 0$:

$$\left.\frac{dET_{avg}}{dx}\right|_{at \; x=0} = \frac{1}{\left(\left(\frac{P_2}{PET_2}\right)^n + 1\right)^{1+1/n}} - \frac{1}{\left(\left(\frac{P_1}{PET_1}\right)^n + 1\right)^{1+1/n}} \quad . \quad (15)$$





This dimensionless number depends only on the aridity indices P/PET at the two sites, and could be used as a screening tool to find regions where lateral redistribution could potentially be most consequential.

Another benchmark for the potential importance of lateral transfer is the maximum possible average ET rate, if lateral transfer took place at its optimal value $x_{opt}$. This quantity can be calculated by substituting the optimal transfer rate $x_{opt}$ (Eq. 14) into our modified Budyko formula (Eq. 11):

$$ET_{opt} = \frac{\frac{P_1 + P_2}{2}}{\left(\left(\frac{P_1 + P_2}{PET_1 + PET_2}\right)^n + 1\right)^{1/n}} \quad . \quad (16)$$

Equation 16 shows that the optimal rate of ET (including lateral redistribution) equals the Budyko curve estimate of ET at the average P and average PET. As shown in the Appendix, this result is quite general, and does not depend on the specific Budyko curve equation that we have used here, nor on any specific number of columns. It requires only that all of the columns are governed by the same downward-curving function in a coordinate space defined by ET/PET and P/PET.

This result demonstrates an interesting connection with the analysis of heterogeneity bias presented above. The maximum possible increase in ET from lateral redistribution exactly equals the heterogeneity bias calculated in the preceding section: both are equal to the ET function at the average P and PET (e.g., Eq. 16 in the case of two columns), minus the average of the ET's calculated for the individual columns using their individual P's and PET's. That is, both are equal to "the function of the averages", minus "the average of the functions". Putting the same point differently, the ET that an Earth system model calculates from average P and PET (the "function of the averages") is not just an overestimate of the true ET (as explained in Section 3 above), it is the _highest possible_ ET under optimal redistribution of the available water.

This observation simplifies the problem of estimating the maximum possible effect of lateral redistribution in heterogeneous terrain: one simply needs to compare the average of the ET's calculated for every pixel within some domain using those pixels' individual P's and PET's, and the ET calculated from the average P and average PET using the same Budyko curve. Alternatively, one can approximate these quantities from the means and variances of P and PET, using Eqs. (6-8).

Of course, any of these estimates of the potential effects of lateral redistribution ignore many real-world constraints, such as topographic or lithologic barriers that could prevent lateral transfer between specific locations (e.g., water will not flow uphill). Thus this estimate should be considered as only a theoretical upper bound.

### 4.3 Real-world example of redistribution effects on estimated ET

To illustrate the possible effects of lateral redistribution on average ET in the real world, we will take as an example of the 1° by 1° grid cell shown in the middle right of Fig. 4a and 5a, which lies at the foot of the Himalayan Front at 89-90° E, 26-27° N. As before, we use 30-arc-second (~1 km) P, PET, and topographic data (from WorldClim, MODIS, and SRTM; (Hijmans et al., 2005; Mu et al., 2007; Jarvis et al., 2008) to represent the finer-scale heterogeneity within this grid cell.





Figure 10 shows three locations that have been selected to illustrate the possible effects of lateral redistribution on average ET. Location 3 is close to sea level, whereas location 2 is at 300 m altitude and location 1 is at roughly 3000 m. We analyzed the effects of a hypothetical redistribution of 500 mm yr$^{-1}$ of water from location 1 to location 2, and from location 2 to location 3.

As Figure 11 shows, P and the aridity index increase dramatically from location 1 (at 3000 m) to location 2 (at 300 m); that is, the landscape becomes more humid as one moves downhill. Using the rule of thumb developed above, one would expect that lateral transfer from location 1 to location 2 should result in a net decrease in average ET. Figure 12a confirms that, as expected, lateral transfer would move the two points farther apart on the Budyko curve, resulting in a net decrease of 9.3 % in the average ET of the two locations.

Conversely, as Figure 11 shows, as one moves downhill from location 2 to location 3, the landscape becomes more arid (the aridity index decreases); thus the rule of thumb outlined above predicts that downhill lateral transfer should result in a net increase in average ET. This expectation is confirmed by Figure 12b; the two locations move closer together on the Budyko curve, resulting in a net 4 % increase in the average ET of the two locations.

## 5 Summary and discussion

The atmosphere mixes and integrates inputs from spatially heterogeneous landscapes. Earth system models average over significant landscape heterogeneity, which can lead to substantial biases in model results if the underlying equations are nonlinear. Due to the mass and energy constraints that limit evapotranspiration rates, ET will be a nonlinear concave-downward function of P and PET, whether expressed by Budyko curves or by other ET models. As a result, ET values calculated from averages of spatially varying P and PET will overestimate the average of the spatially variable ET (the function
of the average will overestimate the average of the function).

In Section 3 above we outlined an approach for estimating this heterogeneity bias, using Budyko curves as a simple empirical ET model. One should keep in mind that Budyko curves are empirically calibrated to catchment-averaged precipitation and discharge (to calculate ET); thus they already average over the spatial heterogeneity within each calibration catchment. This inherent spatial averaging should make Budyko curves smoother (less curved) than the point-scale relationships that determine
ET as a function of P and PET. In other words, the true mechanistic equations that drive point-scale ET may be much more sharply curved than Budyko curves (which already include significant averaging, and thus must plot inside the curve of the raw point-scale data, if such data were available). As a result, the effects of sub-grid heterogeneity and lateral redistribution could potentially be larger than what we have estimated here.

In Section 4, we explored the possibility that lateral transfers of water from one location to another could change the average
ET as seen from the atmosphere. Exploring this question requires a modified Budyko framework, in which one accounts for the water that is available for evapotranspiration (P + net transfer) rather than precipitation alone. This is consistent with Budyko's original approach, which was based on mass balances in catchments with no long-term groundwater gains or losses



(i.e., with no net transfer, and thus with the long-term supply of available water equal to precipitation). Our analysis shows that in regions where the aridity index increases with altitude, lateral redistribution would transfer water from more humid uplands to more arid lowlands, resulting in a net increase in ET (points move closer together on the Budyko curve; Fig. 12b). Alternatively, in regions where the aridity index decreases with altitude, lateral transfer would redistribute water from more arid uplands to more humid lowlands, resulting in a net decrease in average ET (Fig. 12a). We derived simple analytical formulas for estimating the marginal ET effect of a unit of lateral redistribution, as well as the maximum possible ET effect resulting from an optimal (i.e., ET-maximizing) amount of lateral redistribution. Water transfers will most strongly affect average ET if the source (or recipient) location is energy-limited and the recipient (or source) location is water-limited.

Our analysis of redistribution effects is based on the assumption that lateral transfers will reduce the available water at the source location by the same amount that they increase it at the receiving location. Thus we are assuming that water that is redistributed becomes unavailable for evapotranspiration at the source location (for example, through rapid runoff to channels, or rapid infiltration to deep groundwater via preferential flowpaths). Alternatively, if the redistributed water were assumed to come only from surplus that is "left over" after evapotranspiration, the available water (and thus ET) in the source location would not be reduced while the available water (and thus ET) in the receiving location would be increased. Under that assumption, any redistribution would increase average ET, regardless of the climatic conditions in the source and receiving locations. By assuming that available water is conserved (in the sense that whatever is gained in one location is lost from another), our analysis may underestimate the effect of redistribution on average ET.

It bears emphasis that our analysis of the effect of lateral redistribution is inherently hypothetical. By estimating the ET effect of a (hypothetical) transfer of water from one location to another, we are not implying that such a transfer would actually take place at the assumed rate (or would even occur at all) in the real world. Perhaps in reality there is no flowpath connecting the two locations, for example, or perhaps its conductivity is very low, or perhaps the putative source location lies downhill from the putative recipient location. Likewise, although there may be an aquifer connecting two locations, it may lie too deep below the rooting zone to have any significant impact on evapotranspiration rates. Estimating the potential effects of lateral redistribution on ET in real-world cases (rather than hypothetical ones) will require careful attention to such matters, which are beyond the scope of this paper.

The analysis that we have used to quantify the effects of spatial heterogeneity and redistribution could also be used to study the effects of temporal heterogeneity in water availability for evapotranspiration, and temporal redistribution by storage of groundwater between wet and dry seasons. Temporal heterogeneity (e.g., seasonality) in water availability could substantially affect average ET, particularly in climates that shift seasonally between water-limited and energy-limited conditions. In such cases, ET estimates calculated from time-averaged P and PET will be higher than the average of individual ET estimates derived from daily or monthly values for available water and PET. Similarly, temporal redistribution of available water between water-limited and energy-limited conditions (through, e.g., inter-seasonal groundwater storage) could substantially increase average ET. The formulas and approaches we have outlined above could be straightforwardly applied to quantify these temporal heterogeneity and redistribution effects. If, however, one bases such an analysis on Budyko curves as an ET





model, one should keep in mind that these empirical curves are based on long-term catchment mass balances, and thus they already average over seasonal and shorter-term variations in water availability and PET. Thus Budyko curves may already be substantially smoother (less curved) than the short-term behavior that they average over. As a result, any such analysis based on Budyko curves may underestimate the impact of temporal heterogeneity and redistribution on average ET.

Our analysis does not explicitly account for how changes in ET may affect atmospheric humidity and thus PET. This "complementarity" feedback between ET and PET is potentially important for mechanistic models of the evapotranspiration process, and could potentially change the magnitude (though not the sign) of the ET effects that we have estimated in this paper. Any such changes should be small, however, because Budyko curves are empirical relationships derived from catchment mass balances, which already subsume any feedbacks between ET and PET that arise in the calibration catchments.

The simplicity of the approach presented here is both a limitation and an advantage. On the one hand, this simple approach necessarily overlooks, or implicitly subsumes, many mechanistic relationships that would be explicitly treated in more complex ecohydrological models. On the other hand, it avoids the calibration issues and data constraints that may limit the applicability of these more complex models. Our simple approach also has the advantage of transparency; as Figs. 3, 4e, 6, and 12 show, one can directly visualize how both spatial heterogeneity and lateral redistribution affect average ET, using a simple graphical

framework. This framework leads to relatively simple analytical expressions and rules of thumb that can be used to gauge where, and when, heterogeneity and lateral redistribution effects on ET are likely to be most important.

An obvious next step is to use the framework developed here to make a first-order estimate of the likely effects of spatial heterogeneity and lateral redistribution on ET, as seen from the atmosphere at regional and continental scales. The approach developed here is well suited to this task because it is simple and relatively general, and its data requirements are modest.

Heterogeneity effects on ET can be estimated from the means, variances, and covariance of P and PET, and, as we have shown, the maximum hypothetical effect of lateral redistribution can be obtained directly from the same analysis. Quantifying the likely real-world effects of lateral redistribution will be much more challenging, since it necessarily requires estimating the real-world magnitudes of these lateral redistribution fluxes. Work on quantifying heterogeneity and redistribution effects on ET at regional and continental scales is currently underway and will be the focus of future papers.

## Appendix A: Generality of redistribution results

Here we demonstrate that the optimal redistribution results presented in Section 4.2 are also valid for any number of locations (not just two) and for any downward-curving ET function that can be plotted on the Budyko axes (not just Eq. 1, which was

used to derive Eqs. 12-16 in Section 4.2).





We begin by assuming a set of $N$ locations $i=1...N$, each characterized by rates of precipitation $P_i$ and potential evapotranspiration $PET_i$. In keeping with the analysis of Section 4, we assume that the rate of evapotranspiration at each location depends on its available water $AW_i$,

$$Available\ Water_i = AW_i = P_i \pm net\ transfers \quad , \quad (A1)$$

and specifically on the ratio of available water to PET, which we denote for future convenience as $R_i$,

$$R_i = \frac{AW_i}{PET_i} \qquad . \quad (A2)$$

We also assume that the evapotranspiration rates at all locations follow the same functional dependence on $AW_i/PET_i$, and that this functional relationship (denoted $f$) can be represented on Budyko-type axes, that is,

$$\frac{ET_i}{PET_i} = f\left(\frac{AW_i}{PET_i}\right) = f(R_i) \quad or \quad ET_i = PET_i\ f(R_i) \quad , \quad (A3)$$

We impose no restrictions on the form of the function $f$, except that it must be downward-curving; that is, its second derivative must be negative everywhere.

The first result to be demonstrated is: if moisture is redistributed among multiple locations, the highest possible average rate of ET will be achieved when all locations have the same ratio $R_i=AW_i/PET_i$ (note that this does not require that the $AW_i$ or the $PET_i$ are the same). We begin by assigning all the locations the same $R$ value, which we denote $R_{opt}$ (recognizing that its optimality is not yet proven). We then show that any further redistribution of an amount of water $y$ from any location $j$ to any other location $k$ (such that $R_j<R_{opt}$ and $R_k>R_{opt}$) will necessarily lead to a decrease in overall ET. The transfer of $y$ from location $j$ to location $k$ implies that

$$R_j = R_{opt} - \frac{y}{PET_j} \quad , \quad \frac{dR_j}{dy} = -\frac{1}{PET_j} \qquad (A4)$$

and

$$R_k = R_{opt} + \frac{y}{PET_k} \quad , \quad \frac{dR_k}{dy} = \frac{1}{PET_k} \qquad . \quad (A5)$$





Taking the second-order Taylor expansion of Eq. (A3), one obtains for $ET_j$:

$$ET_j = PET_j\, f(R_{opt}) + PET_j\, \frac{df}{dR}\frac{dR}{dy}\, y + \frac{PET_j}{2}\frac{d^2f}{dR^2}\left(\frac{dR}{dy}\right)^2 y^2 + \cdots$$

$$= PET_j\, f(R_{opt}) + PET_j\, \frac{df}{dR}\frac{-1}{PET_j}\, y + \frac{PET_j}{2}\frac{d^2f}{dR^2}\left(\frac{-1}{PET_j}\right)^2 y^2 + \cdots$$

$$= PET_j\, f(R_{opt}) - y\,\frac{df}{dR} + \frac{y^2}{2PET_j}\frac{d^2f}{dR^2} + \ \cdots (A6)$$

and similarly for $ET_k$:

$$ET_k = PET_k\, f(R_{opt}) + y\,\frac{df}{dR} + \frac{y^2}{2PET_k}\frac{d^2f}{dR^2} + \ \cdots (A7)$$

Thus the net change in total ET for locations $j$ and $k$ together is

$$\left(ET_j + ET_k\right) - [PET_j\, f(R_{opt}) + PET_k\, f(R_{opt})] = y^2\left(\frac{1}{2PET_j} + \frac{1}{2PET_k}\right)\frac{d^2f}{dR^2} + \ \cdots (A8)$$

Because the second derivative of $f$ is always negative, the right-hand side of Eq. (A8) will likewise be negative, implying a net decrease in ET for locations $j$ and $k$ whenever $y$ is not zero. The stipulation that the second derivative of $f$ is negative *everywhere* guarantees that any higher-order terms that have been omitted from the Taylor expansion must be too small to change the sign of the right-hand side of Eq. (A8).

Thus the general result is demonstrated for the individual pair of locations $j$ and $k$. Demonstrating that this result is true for this pair of locations is sufficient to prove the general case, since any pattern of water redistribution among any combination of locations is equivalent to a linear combination of such pairwise water transfers.

The second result to be demonstrated is that, for any Budyko-type function $f$ and any combination of locations, the optimal rate of ET (including lateral redistribution among the locations) will equal the Budyko curve estimate at the average P and average PET. For a set of locations $i$, Eq. (A3) implies an average ET of




$$\overline{ET} = \overline{PET_i\ f(R_i)} \quad , \quad (A9)$$

where overbars indicate averages over all locations. As demonstrated above, under optimal redistribution each location will have $R_i=R_{opt}$, such that Eq. (A9) becomes

$$\overline{ET}_{opt} = \overline{PET}\ f(R_{opt}) \quad . \quad (A10)$$

What remains to be proven is that $R_{opt} = \overline{P}/\overline{PET}$ . If we denote the net transfer of water into each location as $z_i$ (such that locations that have a net gain of available water have $z_i>0$, and locations that have a net loss of available water have $z_i<0$), for each location we can write

$$R_i = \frac{AW_i}{PET_i} = \frac{P_i + z_i}{PET_i} \quad \text{or} \quad R_i\ PET_i = P_i + z_i \quad . \quad (A11)$$

Summing Eq. (A11) over all locations, noting that under any mass-conserving redistribution the $z_i$'s must sum to zero and under optimal redistribution $R_i=R_{opt}$ everywhere, we directly obtain

$$\sum R_i\ PET_i = R_{opt} \sum PET_i = \sum P_i + z_i = \sum P_i \quad (A12)$$

and therefore

$$R_{opt} = \frac{\sum P_i}{\sum PET_i} = \frac{\overline{P}}{\overline{PET}} \quad . \quad (A13)$$

Combining Eqs. (A10) and (A13), we have

$$\overline{ET}_{opt} = \overline{PET}\ f\left(\frac{\overline{P}}{\overline{PET}}\right) \quad , \quad (A14)$$

thus proving the second general proposition.



**Author contribution**

Both authors have contributed equally to all aspect of this work.

**Competing interests**

The authors declare that they have no conflict of interest.

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





**Tables:**

| Equation | Parameter | Reference |
|---|---|---|
| $$\frac{ET}{PET} = \frac{P}{PET} \frac{1}{\left(\left(\frac{P}{PET}\right)^n + 1\right)^{1/n}} \quad (1)$$ | $n$ (dimensionless) modifies the partitioning of P between E and Q | Bagrov, 1953; Turc, 1954; Mezentsev, 1955; Pike, 1964; Choudhury, 1999; Zhang et al., 2001; Milly and Dunne, 2002; Yang et al., 2008 |
| $$\frac{ET}{PET} = \frac{P}{PET} + 1 - \left(\left(\frac{P}{PET}\right)^\omega + 1\right)^{1/\omega} \quad (2)$$ | $\omega$ – similar to n, modifies the partitioning of P between E and Q | Fu, 1981; Zhang et al., 2004; Yang et al., 2007 |
| $$\frac{ET}{PET} = \frac{\frac{P}{PET} + \omega}{1 + \omega \left(\frac{P}{PET}\right)^{-1} + \frac{P}{PET}} \quad (3)$$ | $\omega$ – coefficient of vegetation and water supply | Zhang et al., 2001 |
| $$\frac{ET}{PET} = \frac{P}{PET} \frac{exp\left[\gamma\left(1 - \frac{P}{PET}\right)\right]^{-1}}{exp\left[\gamma\left(1 - \frac{P}{PET}\right)\right] - \frac{P}{PET}} \quad (4)$$ | $\gamma$ – the ratio of soil water storage capacity to precipitation | Milly, 1993; Porporato et al., 2004 |

**Table 1. Alternative empirical equations for mean annual evaporation rate in Budyko framework: ET is mean annual evapotranspation, P is mean annual precipitation, PET is mean annual potential evapotranspiration (evaporative demand).**




**Figures:**

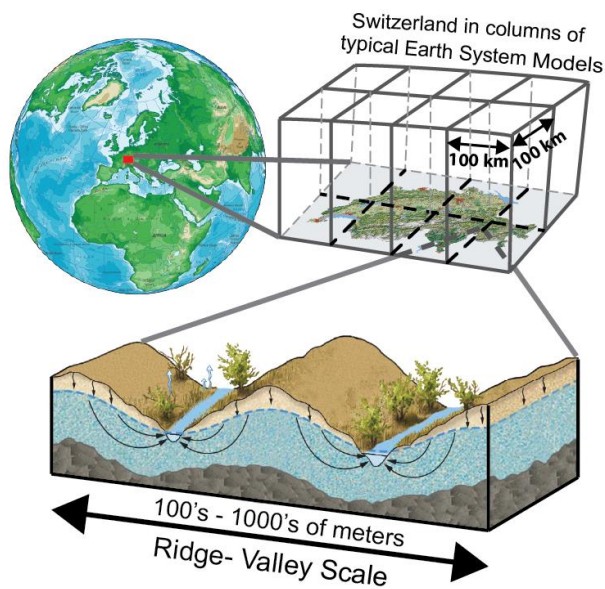

**Figure 1. Sub-grid-scale surface heterogeneity and subsurface water redistribution are unrepresented in Earth System Models. At the 100 km by 100 km grid cell scale, large mountain ranges (such as the Swiss Alps) become indistict.**

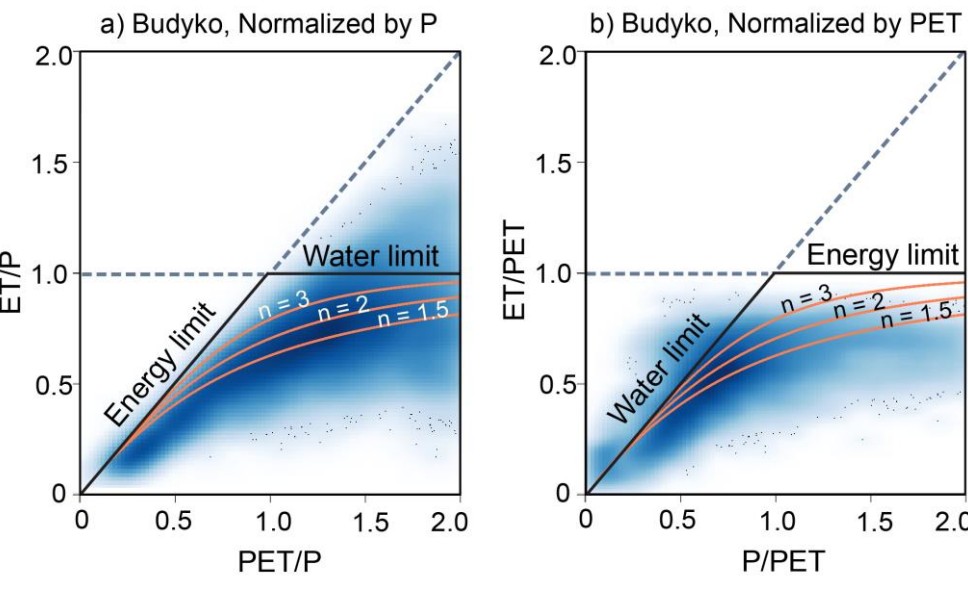

**Figure 2. Budyko framework and energy and water limit lines. The blue cloud is a smoothed scatterplot of the 30 arc-seconds resolution mean annual precipitation (P), evapotranspiration (ET), and potential evapotranspiration (PET) for continental Europe.**
10  **ET and PET data are from MODIS (Mu et al., 2007, P dataset is from WorldClim (Hijmans et al., 2005).**





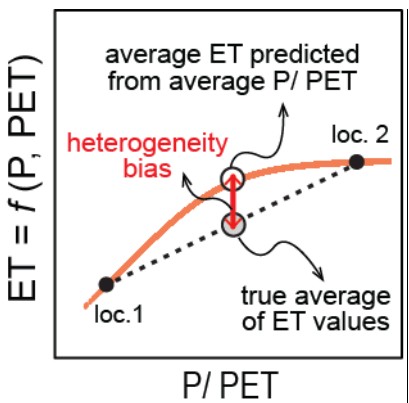

**Figure 3. Illustration of heterogeneity bias in Budyko curve (Eq. 5). The true average (gray circle) of the ET values of locations 1 and 2 (black dots) is less than the average ET that would be estimated from their average P/PET (open circle). The size of the heterogeneity bias will be proportional to the curvature in the ET function and proportional to the variability in P and PET among 10 the individual points (Eqs. 6-8).**





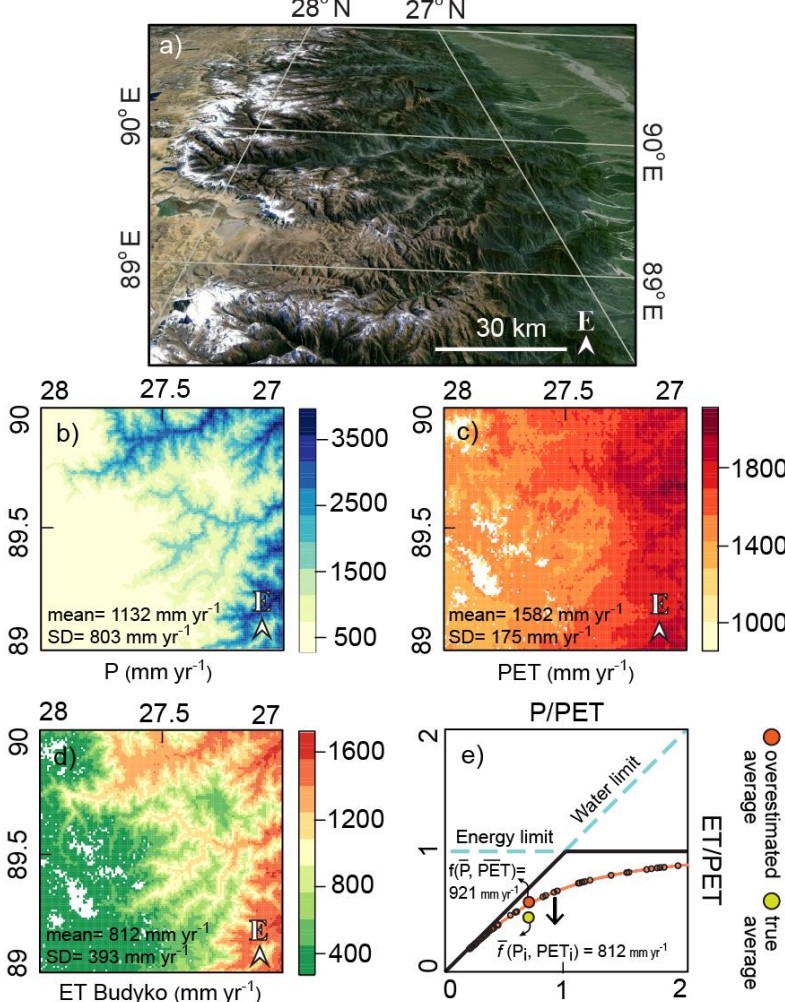

**Figure 4.** One-kilometer topography (a: SRTM, (Jarvis et al., 2008) and annual mean climatology for a 1° by 1° grid cell spanning the Himalayan Front at 89-90° E, 27-28° N. Spatial patterns of 1-km resolution mean annual precipitation (b: WorldClim, (Hijmans et al., 2005), potential evapotranspiration (c: MODIS, (Mu et al., 2007), and (d) evapotranspiration (ET) calculated using the Budyko curve (Eq. 5). (e) shows a random sample of 50 points from (b), (c), and (d), along with the average P, PET, and ET over the grid cell (yellow circle), and the ET value estimated from Eq. (5) for the same average P and PET (orange circle). This ET estimate is 921 mm yr⁻¹, 11.8 % more than the average of the 1-km resolution ET estimates.





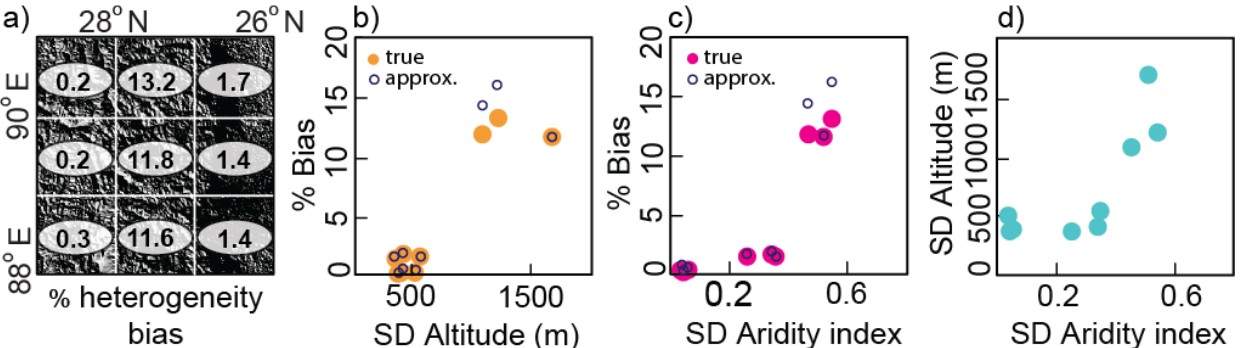

Figure 5. Heterogeneity bias in average ET for the 9 grid cells of the terrain shown in Fig. 4a (88-91° E, 26-29°N) calculated from high-resolution (1 km) spatial variation of mean annual P and PET in each grid cell (Eq. 7). "True" heterogeneity bias is estimated by averaging the ET predicted by the Budyko curve for each 1-km pixel, and comparing this average with the ET predicted from the same curve using the average P and PET in the corresponding grid cell. "Approximate" heterogeneity bias is estimated from Eq.(8). The % bias is highest in cells with large standard deviation in altitude and aridity index.

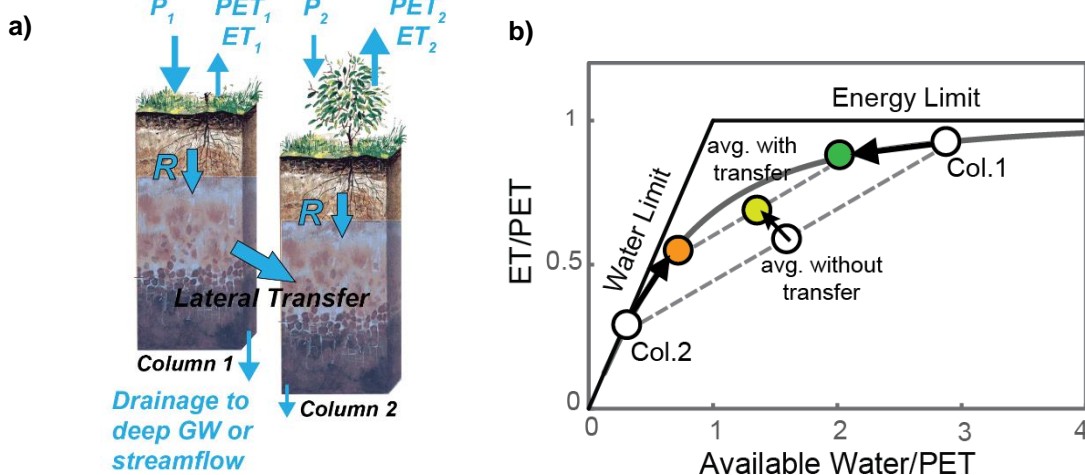

Figure 6. a) A conceptual two-column model. b) Illustration of how the two points representing the two columns shift towards each other in Budyko space if water is transferred from the upper, wetter column to the lower, drier column. Open circles represent columns without lateral transfer and solid circles represent columns with lateral transfer.



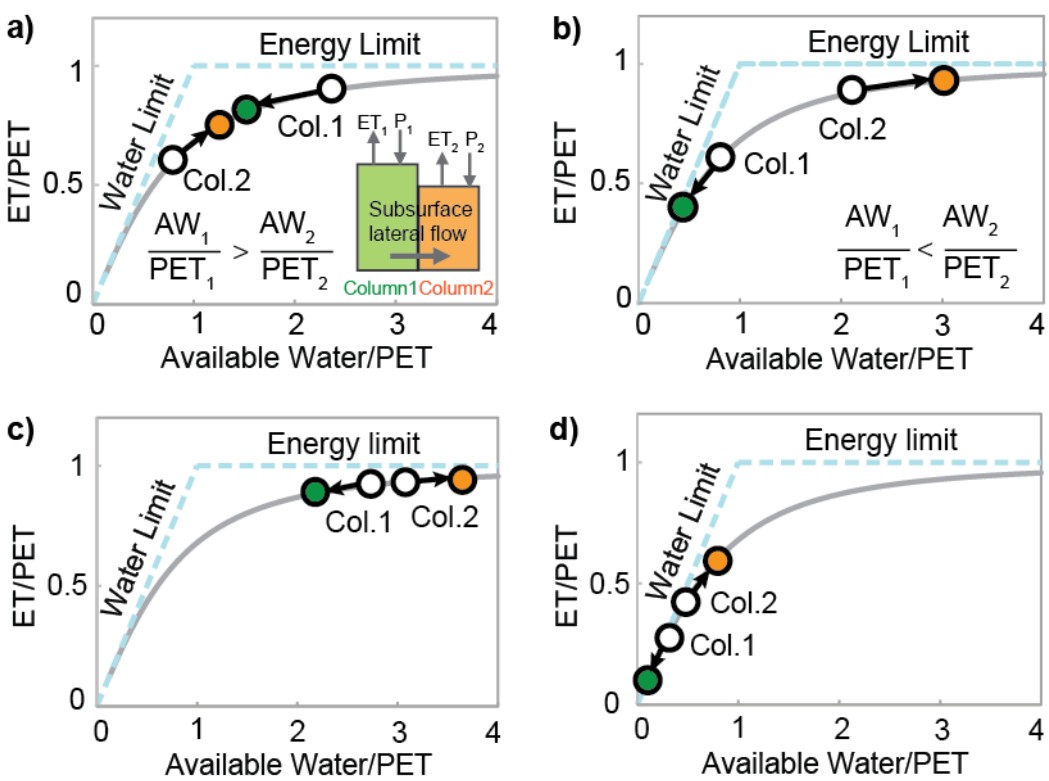

**Figure 7. Four conceptual cases in two-column model where column 1 is topographically always higher than column 2 (water always moves from column 1 to column 2). Open circles represent columns without lateral transfer and solid circles represent columns with lateral transfer. Depending on the columns' wetness or dryness ( P and PET), lateral transfer can potentially a) increase average ET (the points representing column 1 and column 2 are pushed towards one another, spanning significant curvature in the ET function), b) decrease average ET (points are pushed away from one another, spanning significant curvature in the ET function), or (c and d) have little effect on average ET (the columns shift almost collinearly along the energy-limit or water-limit limbs of the curve).**





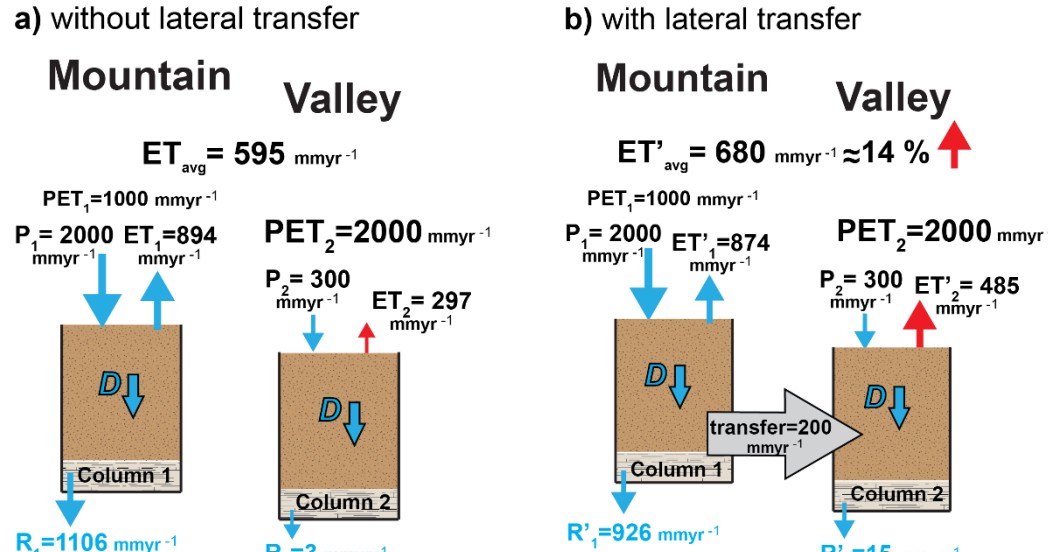

**Figure 8. Hypothetical numerical experiment with conceptual two-column model, a) no lateral transfer between columns b) 200 mm yr$^{-1}$ lateral transfer from column 1 (mountain) to column 2 (valley) increases average ET by 14 %. The magnitude of P (precipitation), PET (potential evapotranspiration), ET (actual evapotranspiration), and R (recharge) are hypothetical and D is drainage to deep groundwater or streamflow.**

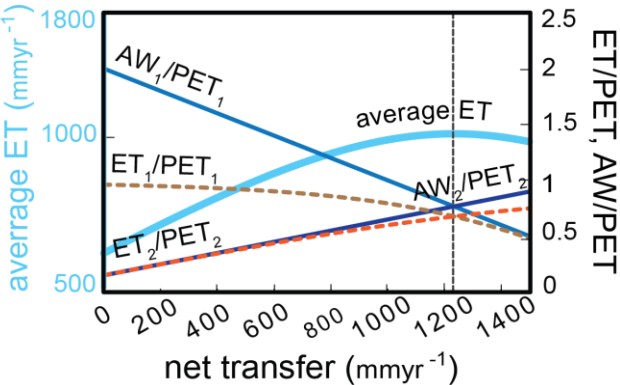

**Figure 9. Average ET is maximized for the rate of net transfer at which P/PET and ET/PET of the two hypothetical columns of Fig. 8 cross one another.**





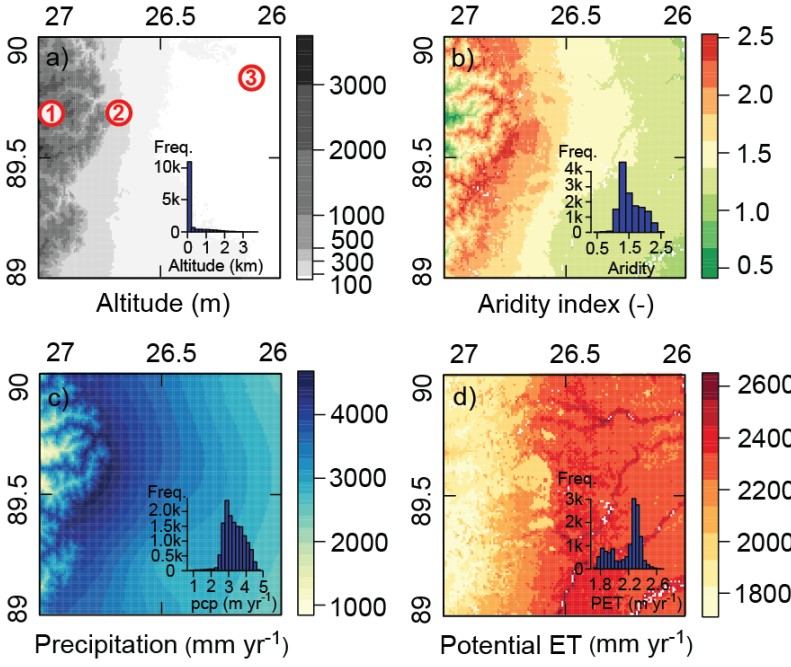

**Figure 10. Spatial patterns of altitude, precipitation (P), potential evapotranspiration (PET) and aridity index (P/PET) in 1° by 1° grid cell in Himalayas at 89-90° E, 26-27° N. There is a sharp gradient in P, PET, and altitude in this grid cell. The labeled points 1, 2, and 3 correspond to the labeled points in Fig. 11.**

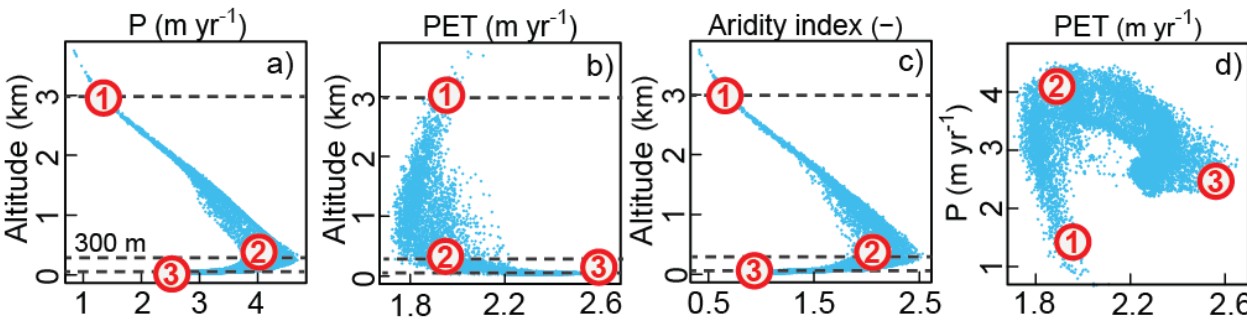

**Figure 11. Variation of precipitation (P), potential evapotranspiration (PET), and aridity index (P/PET) with altitude in 1° by 1°**
10 **grid cell of Himalayas in the extent of Fig. 4. (89-90° E, 26-27° N). P and PET for sites 1, 2 and 3 in Fig. 10 are marked in the graphs. Between locations 3 and 2, P and Aridity index increase and PET decreases with altitude. Between points 2 and 1, P and aridity index sharply decrease and PET slightly increases with altitude.**





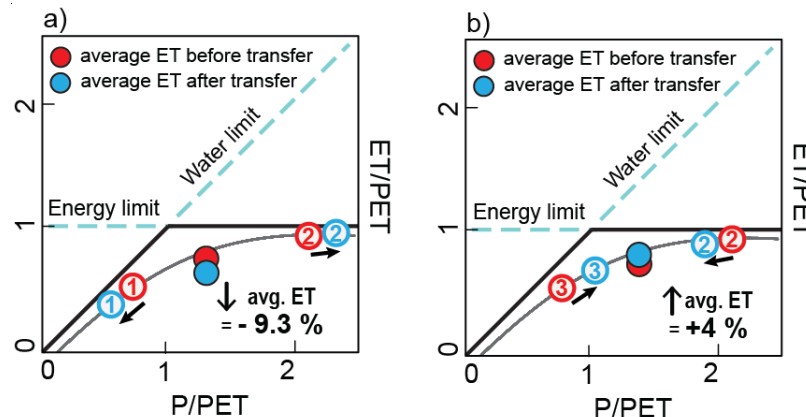

**Figure 12. Budyko curve and increase or decrease of average ET when transfer of water from higher location to lower location is included. a) 500 mm yr⁻¹ of transfer from site 1 (3000 m altitude, lower aridity index) to site 2 (300 m altitude, higher aridity index) decreases the average ET by 9.3 %. b) 500 mm yr⁻¹ of transfer of water from location 2 (altitude 300 m, higher aridity index) to location 3 (altitude 10 m, lower aridity index) increases the average ET by 4 %.**