# Peer review of "A Budyko framework for estimating how spatial heterogeneity and lateral moisture redistribution affect average evapotranspiration rates as seen from the atmosphere"

_Hydrology and Earth System Sciences, 2016_

## Referee Comment (RC1) · Anonymous Referee #1 · 27 Sep 2016

General comments

This manuscript evaluates the effect of averaging the non-linearity in Budyko curve in the context of subgrid heterogeneity in land surface models. Some theoretical results are developed based on the Turc-Mezentsev curve, which relate the overestimation of evapotranspiration as a function of climate variability and co-variability. The next section deals with redistribution of water fluxes into different land surface components. From the previous results, the flux transfer that optimises ET is derived, and this is

further explored using data from the Himalayas. The paper is very clearly written and contains some interesting results. My main criticism of this work is acknowledged by the authors in the discussion, namely that the results are "inherently hypothetical". I was curious as to how the optimal flux transfer is related to actual redistribution fluxes in different landscapes, and whether an "optimality" argument could be explored, i.e. do landscapes naturally organise themselves to generate an optimal flux or not? These questions are however both probably outside the scope of the paper.

Specific comments

I was surprised to not see a reference to Beven (1995), Hydrol. Processes, 9, 507–525, which seems to me to be quite relevant to the current manuscript.

p.5, 5: The use of the term "method of moments" is unfortunate here as this means something quite different (fitting a distribution by matching the first few moments to observed statistics). Really equation (6) is a 2nd order Taylor expansion of the function f about the mean values (as described accurately in the appendix).

P9, 13: I'm not sure what "harmonic difference" means in this context and this is not a term I have come across before. Consider removing this and just refer to "solving eq. 13"

The figures are of a high standard, but seem quite reliant on colour. This may be an issue for print versions of the article?

Technical corrections

p.1, 22 & 24: traditionally the aridity index refers to PET/P rather than P/PET ("humidity index"?). This use of P/PET is discussed in the article, but the use of aridity index for P/PET is potentially confusing, especially in the abstract.

p.2, 18: extra comma and space in the citation here.

There seem to be a few typos in the reference list, e.g. the first entry. Also, I couldn't

find reference to the first author of this first reference (i.e. shouldn't this be Ang and Tang?)

---

## Author Comment (AC1) · 28 Sep 2016

General concern:

Anonymous Referee # 1 pointed out the "inherently hypothetical" character of the moisture redistribution section of our paper, as we ourselves noted in the manuscript (P12, L18). The question of how real-world lateral redistribution fluxes might compare to the optimal fluxes that are calculated in our analysis is of course interesting, as the referee

notes. However, for the reasons that we state in the paper, actual rates of lateral redistribution in the real world remain highly speculative. Thus we see no practical way to determine whether "landscapes naturally organise themselves to generate an optimal flux or not", as the referee puts it. We see no reason to argue that they do so, or that they should. In any event, as the reviewer notes, these questions are outside the scope of the current paper.

Specific comments:

- The Beven (1995) paper is a useful treatment of the sub-grid closure problem in hydrology and can be added to the manuscript, although the manuscript does not concern hydrological modeling per se.

- Method of moments: there appear to be different understandings of this term, so to avoid any confusion, we will instead use the more technical terminology: second-order, second-moment error propagation.

- Harmonic difference: here we knowingly coined a phrase, modeled after the well known harmonic average (that is, the reciprocal of the average of reciprocals). However we can remove it in the interests of simplicity.

- Use of color in figures: HESS prints figures in color so this should not present a problem.

Technical correction:

- P/PET as aridity index: the use of "aridity index" to describe P/PET has been standard terminology in the hydrology and atmospheric science communities ever since UNEP (1992). We agree that this ratio is more properly characterized as a humidity index, as high values of P/PET characterize conditions of high humidity, not high aridity. Nonetheless, the term "aridity index" is more widely used than "humidity index" to describe P/PET.

- Reference and citation formatting issues: we evidently had some difficulties with our

bibliographic software. These glitches will be fixed.

Reference: UNEP: 1992, World Atlas of Desertification. Edward Arnold, London.

---

## Referee Comment (RC2) · M. Roderick (Referee) · 30 Sep 2016

**Review of HESS Manuscript # hess-2016-424**

| Title: | A Budyko framework for estimating how spatial heterogeneity and lateral moisture redistribution affect average evapotranspiration rates as seen from the atmosphere |
|---|---|

| Authors: | E. Rouholahnejad and J. W. Kirchner |
|---|---|

This paper describes how averaging of P & PET does not lead to an average E. The example used is within-catchment lateral transfers of water.

The paper is very well written. For the first time in many years, I did not locate any suggested correction. (I tried hard but could not find anything.) The presentation is a credit to the authors.

The paper is an interesting analysis on an important topic. One important contribution is that the authors present (analytical) estimates for the maximum effect.

I only have two substantive comments.

Page 3, lines 17-22. The authors use PET here as a generic descriptor of demand. Unfortunately, in Hydrologic practice, PET can mean many things (e.g. Penman Open Water, FAO-56, etc…). However, in the original scheme, Budyko actually used net irradiance (sometimes called available energy). It might be useful to point this out here.

Page 12, lines 26-28. A directly analogous result, based on almost identical logic, for averaging over time has been presented previously in HESS (Lim & Roderick, 2015 HESS 18, 31-45).

p. 13, line 3. A great example of this averaging can be found in Gerrits et al (2009, WRR, Vol. 45)

Michael L. Roderick
Australian National University
30/9/2016

---

## Author Comment (AC2) · 1 Oct 2016

General comment

We thank the Referee # 2, Michael Roderick for his comments on the manuscript.

Specific comments

- PET in Budyko framework: In our manuscript we use PET as a generic descriptor

of atmospheric evaporative demand, independent of how it is quantified. However, as Michael Roderick suggests, it is noteworthy to mention that PET could be estimated in many different ways, and to mention that Budyko estimated PET from net radiation.

- Temporal averaging error, Lim and Roderick, 2015, Gerrits et al. 2009: We appreciate Prof. Roderick's mention of these two papers, and it is interesting to see how the same logic holds true using different ET models. We will consider how these points can be incorporated into the text.

---

## Referee Comment (RC3) · Anonymous Referee #3 · 12 Oct 2016

This is a great little paper where the Budyko framework is used together with a second-order Taylor analysis to show that average surface parameters (P and PET) will yield to an overestimation of true evaporation. Next, a simple connected column model is used to show that lateral redistribution of water can either increase or decrease average evaporation. What is really interesting is that the maximum evaporation reached by lateral redistribution is exactly equal to the (positively biased) evaporation. In hindsight this is logical if one realizes that if all the available water P and energy PET is

redistributed over an area we arrive at the evaporation belonging to average P and PET.

The paper is really well written and can deserves publication almost in its present form. There is however one issue that the authors could discuss and one point of partial disagreement.

First, I am not convinced that it safe to assume that redistribution would mean that all the water that is laterally moved to other areas is in available for evaporation. Apart from the fact that the lateral movement is constraint by P-ET (which could I think be build in their approach), the lateral movement of water will almost always happens as either surface flow or saturated (shallow) subsurface flow. This means that this additional water is most likely captured by a stream and lost from the system and thus not all available for lower places. This would mean that the receiving location would move a bit less in the direction of the energy limited domain then in the current model and areal evaporation would end up a bit lower in case of redistribution.

Second, The point of partial disagreement that I have is that a Taylor approach could also be used for temporal averaging. Yes, this could be done, but only if the time scale is such that storage changes can be ignored.

---

## Author Comment (AC3) · 20 Oct 2016

Hydrol. Earth Syst. Sci. hess-2016-424

Response to Referee #3

General comment We thank Referee # 3, for her/his comments on the manuscript.

Specific comments

[Figure]

- First concern We share the reviewer's concern about redistribution assumptions. In the manuscript we have tried to be very clear (page 12, lines 18-25) that our analysis of redistribution effects is inherently hypothetical; it is a "what if" analysis, not a prediction of how much redistribution will actually occur. The manuscript presents examples showing why, in the real world, the redistribution may not occur in the ways that we assumed it would, and we clearly emphasize that estimating the potential effects of lateral redistribution on ET in real-world cases are beyond the scope of this paper.

The reviewer's first point is that lateral movement is constrained by P-ET; that is, that only water that is "left over" after evaporative losses is available for redistribution. Many hydrologists will naturally adopt this as a starting assumption, or even as a simple statement of fact. But in reality all hydrological partitioning results from a competition between ET and gravitational drainage (to deep groundwater or streams), and it is not clear that ET always wins, or that ET's demands are always filled first, particularly when precipitation is seasonal or episodic. Precipitation that drains to great depth, or to streams, before it can be transpired becomes unavailable for evapotranspiration. Thus although it is conventional to think of Q+GW (discharge and net groundwater recharge) as being constrained by P-ET, it is more physically accurate to say that ET, Q, and GW are all constrained by water availability, which in turn is constrained by mass balance (P-ET-Q-GW).

Our analysis of redistribution effects assumes that lateral transfers will reduce the available water at the source location by the same amount that they increase it at the receiving location (Page 12, L9-12). We make this assumption because it is the most conservative, in the sense that it minimizes the net effect of a given amount lateral redistribution on average evapotranspiration.

On page 12, line 11, we discuss the case that the reviewer mentions: if the redistributed water were assumed to come only from surplus that is "left over" after evapotranspiration, the available water (and thus ET) in the source location would not be reduced while the available water (and thus ET) in the receiving location would be increased. As

we point out in the manuscript, that scenario would lead to larger redistribution effects on average ET.

The reviewer also points out that lateral subsurface flows are likely to be captured by streams and thus unavailable for evapotranspiration at other locations. Of course we agree, but we have carefully defined "lateral redistribution", for the purposes of our paper, not as all lateral subsurface flow (regardless of its ultimate fate), but rather as water that does become available for ET elsewhere (either as groundwater flow, or as streamflow that re-infiltrates into valley aquifers). This is obviously only a fraction of all groundwater flow, as the reviewer points out.

- Second concern We agree that there is not a simple analogy between averaging over spatial heterogeneity and averaging over temporal heterogeneity, for the simple reason that the Budyko approach only makes sense over time scales for which storage changes can be ignored. We will make this clear in the revised manuscript. Greve et al. (1) have explored the possibility of using Budyko under unsteady conditions by modifying the Budyko framework, and we will mention this in the text.

Reference: 1. P. Greve, L. Gudmundsson, B. Orlowsky, S. I. Seneviratne, A two-parameter Budyko function to represent conditions under which evapotranspiration exceeds precipitation. Hydrol. Earth Syst. Sci. 20, 2195-2205 (2016).
* * *

---

## Author Response (AR1)

**Point by point reply to reviewers**

\*Note: The page and line numbers correspond to the revised manuscript "hess-2016-424-manuscript-version2.pdf".

**Response to Referee #1**

**General concern:**

Anonymous Referee # 1 pointed out the "inherently hypothetical" character of the moisture redistribution section of our paper, as we ourselves noted in the manuscript (P12, L20). The question of how real-world lateral redistribution fluxes might compare to the optimal fluxes that are calculated in our analysis is of course interesting, as the referee notes. However, for the reasons that we state in the paper, actual rates of lateral redistribution in the real world remain highly speculative. Thus we see no practical way to determine whether "landscapes naturally organise themselves to generate an optimal flux or not", as the referee puts it. We see no reason to argue that they do so, or that they should. In any event, as the reviewer notes, these questions are outside the scope of the current paper.

**Specific comments**

- The Beven (1995) paper is a useful treatment of the sub-grid closure problem in hydrology and is added to the manuscript (page 2, L16-19).

- Method of moments: there appear to be different understandings of this term, so to avoid any confusion, we instead use the more technical terminology: second-order, second-moment error propagation (page 5, L8-9).

- Harmonic difference: here we knowingly coined a phrase, modeled after the well known harmonic average (that is, the reciprocal of the average of reciprocals). However we removed it in the interests of simplicity (page 9, L16).

- Use of color in figures: HESS prints figures in color so this should not present a problem.

**Technical correction:**

- P/PET as aridity index: the use of "aridity index" to describe P/PET has been standard terminology in the hydrology and atmospheric science communities ever since UNEP (1992). We agree that this ratio is more properly characterized as a humidity index, as high values of P/PET characterize conditions of high humidity, not high aridity. Nonetheless, the term "aridity index" is more widely used than "humidity index" to describe P/PET.

- Reference and citation formatting issues: we evidently had some difficulties with our bibliographic software. These glitches are fixed (page 2, L21; page 17, L12).

**Response to Referee #2**

**General comment**

We thank the Referee # 2, Michael Roderick for his comments on the manuscript.

**Specific comments**

**- PET in Budyko framework:**

In our manuscript we use PET as a generic descriptor of atmospheric evaporative demand, independent of how it is quantified. However, as Michael Roderick suggests, it is noteworthy to mention that PET could be estimated in many different ways, and to mention that Budyko estimated PET from net radiation (page 3, L24).

**- Temporal averaging error, Lim and Roderick, 2015, Gerrits et al. 2009:**

We appreciate Prof. Roderick's mention of these two papers, and it is interesting to see how the same logic holds true using different ET models. Lim and Roderick, 2014 is now mentioned in the manuscript (page 13, L2-3).

**Response to Referee #3**

**General comment**

We thank Referee # 3, for her/his comments on the manuscript.

**Specific comments**

**- First concern**

We share the reviewer's concern about redistribution assumptions. In the manuscript we have tried to be very clear (page 12, L20-27) that our analysis of redistribution effects is inherently hypothetical; it is a "what if" analysis, not a prediction of how much redistribution will actually occur. The manuscript presents examples showing why, in the real world, the redistribution may not occur in the ways that we assumed it would, and we clearly emphasize that estimating the potential effects of lateral redistribution on ET in real-world cases are beyond the scope of this paper.

The reviewer's first point is that lateral movement is constrained by P-ET; that is, that only water that is "left over" after evaporative losses is available for redistribution. Many hydrologists will naturally adopt this as a starting assumption, or even as a simple statement of fact. But in reality all hydrological partitioning results from a competition between ET and gravitational drainage (to deep groundwater or streams), and it is not clear that ET always wins, or that ET's demands are always filled first, particularly when precipitation is seasonal or episodic. Precipitation that drains to great depth, or to streams, before it can be transpired becomes unavailable for evapotranspiration. Thus although it is conventional to think of Q+GW (discharge and net groundwater recharge) as being constrained by P-ET, it is more physically accurate to say that ET, Q, and GW are all constrained by water availability, which in turn is constrained by mass balance (P-ET-Q-GW).

Our analysis of redistribution effects assumes that lateral transfers will reduce the available water at the source location by the same amount that they increase it at the receiving location (page 12, L11-14). We make this assumption because it is the most conservative, in the sense that it minimizes the net effect of a given amount lateral redistribution on average evapotranspiration.

On page 12, line 13, we discuss the case that the reviewer mentions: if the redistributed water were assumed to come only from surplus that is "left over" after evapotranspiration, the available water (and thus ET) in the source location would not be reduced while the available water (and thus ET) in the receiving location would be increased. As we point out in the manuscript, that scenario would lead to larger redistribution effects on average ET.

The reviewer also points out that lateral subsurface flows are likely to be captured by streams and thus unavailable for evapotranspiration at other locations. Of course we agree, but we have carefully defined "lateral redistribution", for the purposes of our paper, not as all lateral subsurface flow (regardless of its ultimate fate), but rather as water that *does* become available for ET elsewhere (either as groundwater flow, or as streamflow that re-infiltrates into valley aquifers). This is obviously only a fraction of all groundwater flow, as the reviewer points out.

**- Second concern**

We agree that there is not a simple analogy between averaging over spatial heterogeneity and averaging over temporal heterogeneity, for the simple reason that the Budyko approach only makes sense over time scales for which storage changes can be ignored. This is mentioned in the text in page 13, L3-7.

**List of all relevant changes made in the manuscript**

- Page 2, L16-19

- Page 2, L21
- Page2, L30
- Page 3, L24
- Page 4, L8-9
- Page 9, L16
- Page 13, L2-3
- Page 17, L10-11; L16-17
- Page 18, L32-33
- Page 19, L3-4
- Page 22, L10

[revised manuscript text omitted]